# Effect of matcha green tea on cognitive functions and sleep quality in older adults with cognitive decline: A randomized controlled study over 12 months

**Kazuhiko Uchida**[1,2☯]*, **Kohji Meno**[2☯], **Tatsumi Korenaga**[2], **Shan Liu**[2], **Hideaki Suzuki**[2], **Yoshitake Baba**[3], **Chika Tagata**[3], **Yoshiharu Araki**[3], **Shuto Tsunemi**[3], **Kenta Aso**[3], **Shun Inagaki**[3], **Sae Nakagawa**[3], **Makoto Kobayashi**[3], **Tatsuyuki Kakuma**[4], **Takashi Asada**[5,6], **Miho Ota**[7], **Takanobu Takihara**[3], **Tetsuaki Arai**[7]

1 Institute of Biomedical Research, MCBI Inc., Tsukuba, Ibaraki, Japan, 2 Research Division, MCBI Inc., Tsukuba, Ibaraki, Japan, 3 Central Research Institute, ITO EN, LTD., Makinohara, Shizuoka, Japan, 4 Biostatistics Center, Kurume University Graduate School of Medicine, Kurume, Japan, 5 Memory Clinic Toride, Toride, Ibaraki, Japan, 6 Tokyo Medical and Dental University, Bunkyo-ku, Tokyo, Japan, 7 Institute of Medicine, University of Tsukuba, Tsukuba, Ibaraki, Japan

☯ These authors contributed equally to this work.
* kazuhiko.uchida@cbiri.org

**Data Availability Statement:** The data can be shared publicly without permission from a public

## Abstract

### Objective

Lifestyle habits after middle age significantly impact the maintenance of cognitive function in older adults. Nutritional intake is closely related to lifestyle habits; therefore, nutrition is a pivotal factor in the prevention of dementia in the preclinical stages. Matcha green tea powder (matcha), which contains epigallocatechin gallate, theanine, and caffeine, has beneficial effects on cognitive function and mood. We conducted a randomized, double-blind, placebo-controlled clinical study over 12 months to examine the effect of matcha on cognitive function and sleep quality.

### Methods

Ninety-nine participants, including 64 with subjective cognitive decline and 35 with mild cognitive impairment were randomized, with 49 receiving 2 g of matcha and 50 receiving a placebo daily. Participants were stratified based on two factors: age at baseline and *APOE* genotype. Changes in cognitive function and sleep quality were analyzed using a mixed-effects model.

### Results

Matcha consumption led to significant improvements in social acuity score (difference; -1.39, 95% confidence interval; -2.78, 0.002) (*P* = 0.028) as evaluated by the perception of facial emotions in cognitive function. The primary outcomes, that is, Montreal Cognitive Assessment and Alzheimer's Disease Cooperative Study Activity of Daily Living scores,

repository after acceptance. The data files are available from the Dryad Digital Repository. https://doi.org/10.5061/dryad.2280gb61r.

**Funding:** The author(s) received no specific funding for this work.

**Competing interests:** Authors with competing interests Enter competing interest details beginning with this statement: I have read the journal's policy and the authors of this manuscript have the following competing interests: [Kazuhiko Uchida serves as a board member of MCBI Inc. Kohji Meno, Tatsumi Korenaga, Liu Shan, and Hideaki Suzuki are employees of MCBI Inc. Yoshitake Baba, Chika Tagata, Yoshiharu Araki, Shuto Tsunemi, Kenta Aso, Shun Inagaki, Sae Nakagawa, Makoto Kobayashi, and Takanobu Takihara are employees of ITO EN, LTD. This research received no external funding.] This does not alter our adherence to PLOS ONE policies on sharing data and materials.

showed no significant changes with matcha intervention. Meanwhile, Pittsburgh Sleep Quality Index scores indicated a trend toward improvement with a difference of 0.86 (95% confidence interval; -0.002, 1.71) ($P$ = 0.088) between the groups in changes from baseline to 12 months.

## Conclusions

The present study suggests regular consumption of matcha could improve emotional perception and sleep quality in older adults with mild cognitive decline. Given the widespread availability and cultural acceptance of matcha green tea, incorporating it into the daily routine may offer a simple yet effective strategy for cognitive enhancement and dementia prevention.

## Introduction

Dementia, for which aging is the greatest risk factor, is a social burden in an aging society. According to the World Alzheimer Report 2022, the number of dementia patients will increase from 57.4 million in 2019 to 152.8 million in 2050 [1]. Alzheimer's disease (AD) and related disorders have multifactorial and complex etiologies [2]. Despite advancements in the development of therapeutic agents, no curative therapies are currently available.

Accumulating evidence suggests that lifestyle habits after middle age (approximately 45–54 years-old) significantly impact the maintenance of cognitive function in older adults [3–5]. Tea intake is closely related to lifestyle habits. Matcha green tea powder (matcha), which contains epigallocatechin gallate (EGCG), theanine, and caffeine, has the beneficial effects of each constituent on cognitive functions and mood [6–8]. Catechins, key components of green tea, are polyphenols with antioxidant and anti-inflammatory properties. The oral administration of EGCG has been reported to reduce Aβ deposition in the olfactory cortex and hippocampus of AD mice [9]. Theanine has been shown to work synergistically with catechins to alleviate stress and improve sleep in both clinical and animal studies [10]. A cross-sectional observational study revealed that the risk of cognitive decline, indicated by a score of < 26 on the Mini Mental State Examination-Japanese version (MMSE-J), was significantly lower in the group that consumed ≥ 200 ml of green tea per day, indicating a positive correlation between regular green tea consumption and maintained cognitive function [11]. With a cutoff point of 26 in MMSE-J score, the odds ratios for the different frequencies of green tea consumption were 0.42 for 2–3 cups per day, compared with 1.00 (reference) for 3 cups per week. Two to three cups of green tea contain 135–200 mg of catechins, with green tea providing 67.5 mg of catechins per 100 ml [11, 12]. In comparison, in a traditional tea ceremony, Tea Otemae, 2 g of matcha contain 170 mg of catechins.

Randomized, double-blind, placebo-controlled trials have reported beneficial effects of matcha on cognitive function [6, 7, 13]. However, the intervention periods in these studies were relatively short, lasting only 12 weeks. Longer intervention periods, extending several months or more, are required to comprehensively evaluate the effects of matcha consumption as a lifestyle habit.

Aging causes sleep quality to deteriorate. Approximately 20% of adults in Japan experience sleep disturbance [14, 15], and 50–70 million people have sleep disorders in the United States [16]. Sleep disturbance is a risk factor for AD, and sleep deprivation leads to increased

inflammation and amyloid beta (Aβ) burden in the brain [17, 18]. A sleep duration of ≤ 6 h at ages 50 and 60 years, compared with a 7-h sleep duration, was associated with a 30% increased dementia risk in the Whitehall II study [19]. Thus, sleep quality during midlife may contribute to the pathophysiology of cognitive impairment in old age, and improving sleep quality may prevent dementia.

Preventing disease progression and disease onset is crucial for mitigating the prevalence of dementia and other lifestyle-related diseases. Controlling blood pressure and blood sugar levels in midlife is vital for the prevention of cardiovascular disease and diabetes mellitus, respectively. Several cohort studies have identified blood-based biomarkers for dementia risk assessment, including phosphorylated tau-217 [20], α2-macroglobulin [21], apolipoproteins [22], and complement proteins [23]. Indicators of protein nutrition are likely associated with the disease processes in AD [24–26]. Llewellyn *et al.* reported an association between low serum albumin levels and risk of cognitive impairment in older individuals [26]. In our previous studies, a composite marker of apoA1, transthyretin, and complement protein C3 in the serum, which are involved in lipid metabolism, protein nutrition, and inflammation, could differentiate between older adults with cognitive impairment and those without dementia at 74– 84% accuracy [27, 28]. Thus, lifestyle habits and related biomarkers are pivotal factors in the prevention of dementia in the preclinical stages.

Here, we conducted a 12-month randomized, double-blind, and placebo-controlled clinical trial to assess the potential of matcha in improving sleep quality and delaying cognitive decline in older adults.

## Materials and methods

### Study design

A 12-month intervention was executed employing a randomized, double-blind, placebo-controlled, parallel-group design. Before the initiation of the study, the clinical trial was registered with the UMIN Clinical Trials Registry (UMIN-CTR), which is recognized by the International Committee of Medical Journal Editors (ICMJE) as an approved registry. This study was conducted and reported as a randomized controlled trial, adhering strictly to the CONSORT guidelines.

A meta-analysis was conducted to ascertain whether the effect size and statistical power were commensurate with the determined sample size, referencing three intervention studies that employed the Montreal Cognitive Assessment (MoCA) as the evaluative criterion [29– 31]. Although the efficacy indices in this study vary slightly from those in previous research, we concluded that observing the anticipated effects outlined in the meta-analysis would render the data from all 66 patients a significant evidence base for preparing the subsequent validation study.

The required sample size was calculated using G*Power 3 software (Heinrich Heine University, Düsseldorf, Germany). We chose to assess the difference between the medians of two independent groups using the Wilcoxon or Mann–Whitney test. With the parameters set as two-tailed, logistic distribution, an effect size of 0.7, a significance level of 0.05, and a power of 0.8, it was determined that each group required 31 cases. Anticipating a dropout rate of 7 of 40 recruited cases per group, we established a target sample size of at least 33 cases per group to validate the robustness of our findings (details are described in **S1 File**).

### Randomization

**Sequence generation.**   Eligible participants with SCD or MCI were randomized using a computer-generated sequence. Participants were stratified based on two stratification factors,

age at baseline ($\geq$ 74 years / < 74 years) and *APOE* genotype ($\varepsilon$4 positive / negative). Participants were stratified based on an age cut-off of 74-years, because the 10-year AD risk is higher in people $\geq$ 74 years old than in those < 74 years old [32]. Allocation to either the control or intervention group was intended to be randomized in a 1:1 ratio using computerized minimization, which aims to minimize differences between groups with respect to the stratification factors [33].

**Allocation concealment mechanism.**   Participants underwent randomization using a dynamic program implemented via a Microsoft Excel macro, employing simple randomization rather than permuted block randomization. Upon running the program, the participants were assigned to either Group A or Group B. However, the program did not reveal which was the intervention or control group. The randomization table created by the program was retained until all baseline stratification factors were fully aligned and participant inclusion was complete, ensuring concealment of the allocation.

All eligible participants who provided informed consent and met the predefined inclusion criteria underwent randomization, a process administered at the discretion of the study site personnel overseeing recruitment and medical interviews. Subsequently, the study site transmitted a stratification factor response form to an independent allocation manager who was responsible for the allocation process, ensuring separation from the outcome evaluation analysis. The allocation manager utilized a dynamic randomization program incorporating provided stratification factors to assign participants to their respective groups based on the program's output. Physicians and nurses responsible for participant inclusion and symptom assessment remained blinded to group assignments.

The allocation sequence utilized a stratified approach, considering study site and medication compliance, employing a dynamic allocation method with the minimization technique. Concealment of randomization details from data management personnel and statistical analysts was upheld throughout the study while ensuring the database remained publicly accessible. The randomization list was securely maintained by designated allocation personnel throughout the study, mitigating any potential influence from principal investigators involved in the research.

**Blinding.**   Throughout the study duration, participants were unaware of whether the capsules they consumed contained matcha or placebo until the study's conclusion was obtained. Physicians, nurses, and principal investigators at the study site were informed that the participants had been assigned to provisional groups (Groups A and B) yet remained uninformed about the specific allocation to either the matcha or placebo group until the randomization list was released after the study period. No member of the research team knew the allocated sequence until the end of the statistical analysis, after the database was locked.

## Participants

Community-dwelling older adults aged 60–85 years were recruited and assessment for eligibility was performed at the University of Tsukuba Hospital (Tsukuba, Ibaraki, Japan) and Memory Clinic Toride (Toride, Ibaraki, Japan) (**Fig 1**). Participants were enrolled based on predefined inclusion and exclusion criteria as follows: (1) Japanese-speaking men and women aged 60–85 years living with a study partner responsible for checking capsule intake and escorting visitors to the hospital, (2) presence of mild cognitive impairment (MCI) or subjective cognitive decline (SCD) assessed as described in the study design and participants, (3) no history of treatment for any serious disease (cancer, myocardial infarction, stroke) within 5 years; (4) no dementia treatment (donepezil, galantamine, rivastigmine, or memantine); (5) no brain atrophy seen in dementia on magnetic resonance imaging (MRI); (6) no participation in

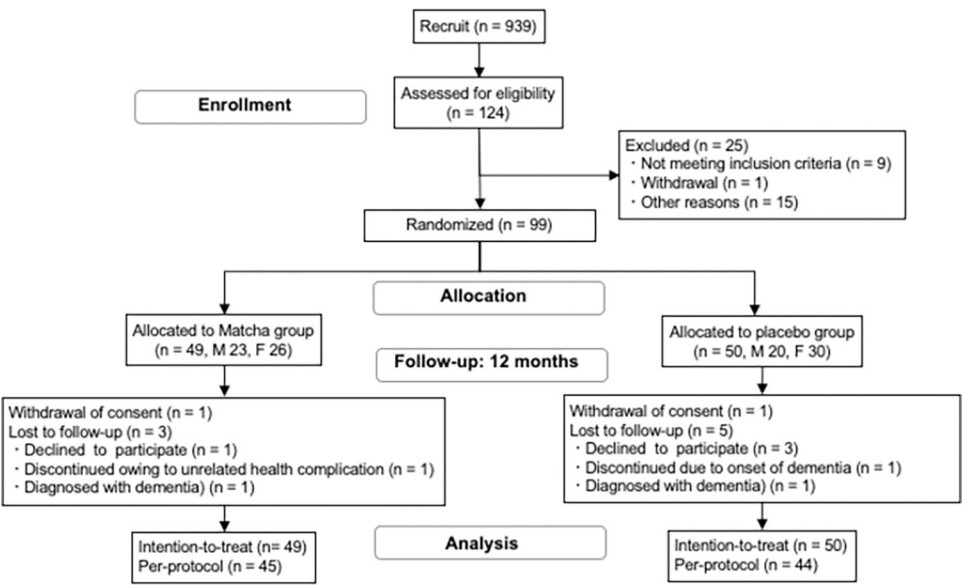

**Fig 1. Study design and flowchart of participants throughout the study.** MCI, mild cognitive impairment; SCD, subjective cognitive decline.

daycare interventions such as exercise intervention or cognitive training; and (7) no history of interventions such as exercise intervention or cognitive training. The exclusion criteria were as follows: (1) diagnosed with dementia or major depressive disorder at baseline; (2) MMSE-J score < 24; (3) history of therapeutic inpatient surgery due to a stroke, subarachnoid hemorrhage, or other head injury; (4) using medications that may affect this study (e.g., antipsychotics, anxiolytics, antidepressants); and (5) regular consumption of functional foods or more than seven cups of matcha per week that may affect the maintenance of cognitive function.

In this study, we included patients with SCD and those with MCI to evaluate the effects of matcha consumption in the pre-stage of cognitive impairment. Although SCD is not classified as a disease, an increasing number of studies have suggested an association between SCD and an elevated risk of developing cognitive impairment [34, 35].

Participants with SCD and MCI were randomly divided into matcha and placebo groups (see Materials and Methods in **S1 File**). No member of the research team knew the allocated sequence until the end of the statistical analysis, after the database was locked. Clinical information, including neuropsychological test data, was collected, and managed using the Alliance Clinical Research Supporting System (ACReSS) V01L51 provided by the University Hospital Clinical Trial Alliance (UHCT) in Japan.

Neurocognitive tests were performed before and after the intervention (baseline and 12 months) and 6 months after the intervention. Amyloid PET was performed at the baseline and 12 months after the intervention in 12 participants each in the matcha group and placebo group. Written informed consent was obtained from all the eligible participants. The study was conducted in accordance with the Declaration of Helsinki, and all protocols used in this study were approved by the Institutional Ethics Committee of the University of Tsukuba, Memory Clinic Toride, and MCBI. Prior to inclusion in the study, written informed consent was obtained from each participant, and all samples were rendered anonymous before being sent to the laboratory for analysis. We actively embraced the principles of diversity, equity, and inclusion (DEI) throughout the entirety of our research process. Our commitment to DEI was reflected not only in our team composition but also in our study design, execution, and

interpretation of results. Each participant was treated with respect, dignity, and fairness, and every effort was made to ensure that the research process was culturally competent and responsive to the needs of the diverse participants.

## Procedures for diagnosis

The MCI and SCD criteria were applied to individuals without dementia after the conference. The team reviewed functional, medical, neurological, psychiatric, and neuropsychological data to reach a diagnostic consensus on dementia and MCI according to the DSM-5 (American Psychiatric Association, 2013) criteria. SCD was defined according to recent criteria [36].

Consistent with the standard criteria, for all subtypes of MCI described below, the participants considered to have MCI were required to have (1) objective impairment in one cognitive domain based on the average of scores on neuropsychological measures within that domain, and 1 standard deviation (SD) and 1.5 SD cutoffs derived from normative corrections for age, years of education, and sex; (2) essentially preserved activities of daily living; (3) presence of memory complaints; and (4) no diagnosis of dementia by group consensus.

The participants labeled as having SCD were required to have (1) no diagnosis of MCI or dementia; (2) subjective complaints of forgetfulness; (3) decline in cognitive functioning unrelated to an event explaining the cognitive deficits; (4) Clinical Dementia Rating (CDR) global score 0 and (5) MMSE-J score between 27 and 30, mild hippocampal atrophy on MRI, or mild regional cerebellar blood flow (rCBF) reduction on single-photon emission computed tomography (SPECT).

## Intervention and outcome

The participants were required to consume nine capsules of matcha (equivalent to 2 g of matcha) or placebo daily at any time. As the amount of matcha in the traditional tea ceremony, Tea Otemae, is 2 g, the daily intake quantity of matcha was set at 2 g in this study. The matcha used, "Hojin no shiro" (provided by ITO EN, LTD., Tokyo, Japan) contained 170.8 mg of catechin, 48.1 mg of theanine, and 66.2 mg of caffeine per daily serving. The catechins consisted of 105.3 mg of EGCG, 1.5 mg of gallocatechin gallate, 20.3 mg of epicatechin gallate, 0.1 mg of catechin gallate, 33.1 mg of epigallocatechin, 1.6 mg of gallocatechin, 8.1 mg of epicatechin, and 0.8 mg of catechin. The placebo capsules, which were identical in appearance, color, and odor to the matcha capsules, were composed of cornstarch. Both capsules consisted of white porcine gelatin. A daily supply (nine capsules) was sealed in a sachet, and a large bag containing 1 month's supply of capsules was provided to the participants every 3 months. During the 3-, 6-, 9-, and 12-month visits throughout the intervention periods, intake compliance was assessed by counting the number of capsules that remained in the bag.

Primary outcomes measures were Montreal Cognitive Assessment-Japanese version (MoCA-J) and Alzheimer's Disease Cooperative Study Activity of Daily Living (ADCS-MCI-ADL) scores, and secondary outcome measures were MMSE-J, Repeatable Battery for the Assessment of Neuropsychological Status-Japanese version (RBANS), Alzheimer's Disease Assessment Scale-Cognitive Subscale-Japanese version (ADAS-Jcog), CNS vital signs computerized neurocognitive battery, Cognitrax [37, 38], and the Japanese version of the Pittsburg Sleep Quality Index (PSQI) scores.

## Cognitive function and sleep quality measures

Cognitive function was assessed using MMSE-J, MoCA-J, ADAS-Jcog, RBANS, and CNS vital signs. CNS vital signs contains 6 neurocognitive domain scores assessed by 10 tests. In this study, we used the stroop test and tests for shift attention, continuous performance, and

perception of emotion to evaluate cognitive functioning, based on neurocognitive domain scores of social acuity, reaction time, complex attention, cognitive flexibility, executive function, and simple attention (**S1 Table**). ADCS-ADL-MCI, with 24 items, was used to describe the participants' ADL [39, 40]. Sleep quality was measured by PSQI [41].

## Neuroimaging

Amyloid PET was performed at the AIC Imaging and Inspection Center in Tsukuba, Japan. Scans were performed using a TruePoint Biograph-PET-CT system (Siemens AG, Munich, Germany). 18F-Florbetapir (Fujifilm, Tokyo, Japan) were used as imaging agents. Transmission imaging (CT imaging for attenuation correction) was performed for approximately 30 s, and emission imaging (PET imaging) was performed in ListMode for 20 min. The standard uptake ratio (SUVR) was calculated from acquired PET images.

The voxel-based specific regional analysis system for atrophy detection (VSRAD) approach in MRI allows the evaluation of atrophy in specific regions of interest (ROIs) within medial temporal structures, particularly the entorhinal cortex, hippocampus, and amygdala. Each processed segmented image was compared with the mean and standard deviation of gray matter or white matter images derived from 80 healthy (40 male and 40 female, aged 54 and 86 years) controls. The severity of atrophic changes was calculated using the Z-score, defined as follows: Z-score = ([control mean]–[individual value]) / (control standard deviation). Each voxel had dimensions of $2 \times 2 \times 2$ mm. The severity of ROI atrophy was determined by using the following equation: sum of positive Z-scores in the target ROI/total number of voxels in the target ROI. ROI atrophy was categorized into four states: 0–1 representing normal, 1–2 mild, 2–3 moderate, and > 3 severe, based on the reference literature [42].

The rCBF was evaluated using SPECT. The Z-score also allowed comparisons of the SPECT scans of participants and age-matched normal controls. Statistical analysis of the perfusion reduction was conducted. Severity perfusion reduction in (posterior cingulate gyrus, precuneus, and parietal cortex) was calculated using the following equation: severity = sum of positive Z-scores in the target ROI / total voxels in the target ROI [43].

## Serum and plasma sampling

Serum and EDTA plasma stored at −80˚C were transferred to the laboratory for analysis and thawed on ice; 40-µl aliquots were generated and stored at −80˚C for immunoassay. Almost all serum samples were frozen and thawed twice, whereas some were frozen and thawed three times. Freezing and thawing up to three times did not affect the results of the immunoassay.

## Immunoassay

Plasma Aβ1–40 and Aβ1–42 levels were determined by immunoassay (Euroimmun, Lübeck, Germany). Plasma levels of amyloid precursor protein (APP) and beta-site amyloid precursor protein cleaving enzyme 1 (BACE1), which are involved in Aβ production, were measured using enzyme-linked immunosorbent assay (ELISA). The human sAPPα assay kit (IBL 27734) and human sAPPβ-w assay kit (IBL 27732) were purchased from Immuno-Biological Laboratories (Gumma, Japan), and BACE-1 ELISA (Euroimmun) was used. The immunoassay kit for brain-derived neurotrophic factor was purchased from R&D Systems (Minneapolis, MN). The blood levels of malondialdehyde (MDA)-modified low-density lipoprotein (MDA-LDL), IL-6, and IL-10 were measured at the Clinical Chemistry Laboratory (BIKEN, Osaka, Japan). Plasma levels of apoA1, transthyretin, and complement protein C3 were determined as previously described [28].

## Statistical analyses

The analysis population consisted of all participants, excluding those who met the discontinuation criteria assessed by a physician, discontinued participation in the study, and conformed to the study protocol. We evaluated the effects of the matcha intervention on cognitive function and sleep quality measures using a mixed-effects model that flexibly modeled the variance structure of the effect measures, including intra-individual correlations generated by repeated measurements as follows. Specifically, we employed an unstructured variance that did not presuppose a specific correlation structure and included group effects, timing effects, and group-by-time interaction terms.

$Y_{ijt}$ represents the measured value of the evaluation item at time point $t$ (= 1,2,3) for participant $I$ who was randomly assigned to group $j$ (= 1,2). We assume the following mixed-effects model:

$$Y_{ijt} = \alpha + \beta_j + \gamma_t + \theta_{jt} + \varepsilon_{ijt}$$

$\alpha$ is the intercept, $\beta_j$ is the group effect, $\gamma_t$ is the timing effect, $\theta_{jt}$ is the group and timing interaction, $\varepsilon_{ijt}$ is the random error mean following a normal distribution with zero variance

structure $V(\varepsilon_{ijt}) = \begin{bmatrix} \sigma_1^2 & \sigma_{12} & \sigma_{13} \\ \sigma_{12} & \sigma_2^2 & \sigma_{23} \\ \sigma_{13} & \sigma_{23} & \sigma_3^2 \end{bmatrix}$

The primary hypothesis to be examined is a two-group comparison of the mean change between baseline and time point 3 (and time point 2). We denote this hypothesis as $H : \delta_{t(1-3)} = 0$ $vs$ $K : \delta_{t(1-3)} \neq 0$. Here, $\delta_{t(1-3)}$ is the group difference in the mean change between baseline and time 3, defined as $\delta_{t(1-3)} = (\theta_{11} - \theta_{13}) - (\theta_{21} - \theta_{23})$. The two-group difference in mean change between baseline and time point 2 is defined as $\delta_{t(1-2)}$.

We also estimated the change (mean difference) by time period, $\Delta(BL, 6M) = 6M - baseline$ or $\Delta(BL, 12M) = 12M - baseline$. The difference between groups was defined by the following equation to estimate the difference in means (Estimate) between groups. Estimate ((difference of means A6M − A0M) − (difference of means B6M − B0M)), or Estimate ((difference in means A12M − A0M) − (difference in means B12M − B0M)). A t-test was conducted with the null (H) and alternative (K) hypotheses. $H$: $\delta = 0$, $K$: $\delta \neq 0$. The null hypothesis is "no variation, no intervention effect" and the null hypothesis is rejected, which is "there is an intervention effect". In a mixed-effects model, if an intervention effect is found, there is at least a timing effect and a group effect, and an interaction between timing and group is also found. If $\delta$ is negative, there is an interaction between time and group, which indicates is an intervention effect of matcha intake, meaning higher values in the matcha group.

For biomarker analysis, differences between the intervention and non-intervention groups were tested using the Mann–Whitney U test. The associations between the cognitive tests were analyzed using Spearman's correlation coefficients. $P$-value of $\leq 0.05$ was deemed significant.

## Results

### Participants

As shown in **Fig 1**, we recruited 939 community-dwelling older adults aged 60–85 years and 124 participants were enrolled (from May 30, 2019 to August 31, 2020) according to the inclusion and exclusion criteria at the University of Tsukuba Hospital (Tsukuba, Ibaraki, Japan) and Memory Clinic Toride (Toride, Ibaraki, Japan). A total of 99 participants were randomized and allocated to the matcha group (n = 49) and matched placebo group (n = 50). Age, sex,

and *APOE* genotype were adjusted for. The intervention was initiated from July 2019 to September 2020 and ended from August 2020 to October 2021.

Eight participants discontinued the study due to withdrawal of consent and disease onset, which could have affected continuation of the intervention. Discontinuations due to illnesses included one patient who developed lung cancer, one patient who developed pancreatic cancer, one patient with increased intraocular pressure, one patient with increased transaminases, and two patients with poor physical condition. Two participants withdrew consent.

There were no cases of a causal relationship between harmful events and matcha consumption. The total number of completed questionnaires was 46 in the matcha group and 45 in the placebo group. During the intervention, each participant with MCI in the matcha and placebo groups was diagnosed with dementia at the 12-month evaluation. After excluding these two participants, the per-protocol analysis of the intervention effects included 45 and 44 participants in the matcha and placebo groups, respectively. According to the intention-to-treat principle, we analyzed the changes in outcome variables from baseline to 12 months were statistically analyzed using a mixed-effects model, with 49 and 50 participants in the matcha and placebo groups, respectively.

## Baseline characteristics of participants

All the baseline characteristics of biochemical blood tests and neuroimaging data of the participants, including *APOE* genotype, plasma Aβ, MRI, SPECT, amyloid PET, and Aβ sequester protein [27, 28] levels, are shown in **Tables 1–3 and S2 Table**.

In amyloid PET, participants > 75 years of age and participants with *APOE* ε4 were included regardless of age. MMSE-J, GDS, WMS-R, CDR, MRI, SPECT, and biochemical blood test were performed at baseline. *APOE* genotyping was performed only at baseline, as it was necessary for allocation.

There was no difference in the average age ($P = 0.907$) and percentage of *APOE*4 carriers ($P = 0.876$) between the groups, indicating random assignment. At baseline, all outcomes, including PSQI, cognitive functioning test scores, and ADCS-MCI-ADL scores, showed no difference between the matcha and placebo groups. The average PSQI scores in the matcha and placebo groups were 4.0 and 5.0, respectively, with no significant differences between the groups. Vitamin B12 and folic acid levels at baseline were similar between the groups (**S2 Table**). There were no significant differences in brain atrophy or rCBF at baseline. Amyloid PET was performed on 24 (9 male and 15 female) participants, harboring *APOE*4 genotype at baseline and 12 months, with no significant differences between the groups at baseline (**Table 3**).

## Effect of 12-month matcha consumption on sleep quality and cognitive function

Intake compliance was assessed by counting the number of capsules that remained in the bag at the 3-, 6-, 9-, and 12-month visits. The intake rates were 98–99% throughout the intervention periods. We also verified matcha intake during the intervention period by measuring theanine levels in red blood cells. The matcha group demonstrated a significant increase in theanine levels ($P = 1.9E-04$), which was not observed in the placebo group. This means the participants consumed matcha and placebo capsules correctly and consistently throughout the intervention period.

For the intention-to-treat analysis of each outcome, we tested the intervention effects using mixed-effects models, with 46 and 45 participants in the matcha group and placebo groups, respectively. The model equation consisted of a time effect (baseline, 6 months, and 12

**Table 1. Demographic characteristics of participants in the matcha and placebo groups at baseline.**

| Characteristics | Matcha group | Placebo group |
|---|---|---|
| | (n = 49) | (n = 50) |
| N (M/F) | 49 (23/26) | 50 (20/30) |
| Age | 72.1 ± 6.0* | 71.9 ± 6.1 |
| *APOE* Genotype, ε4, % | 28.6 | 30.0 |
| Aβ1–40, pg/ml | 140.4 (41.1)† | 143.5 (35.3) |
| Aβ1–42, pg/ml | 22.6 (5.6) | 22.9 (6.0) |
| Aβ (42/40) | 0.163 (0.035) | 0.157 (0.037) |
| BACE1, pg/ml | 461.17 (188.69) | 491.93 (211.83) |
| sAPPα, pg/ml | 6.31 (6.94) | 7.75 (6.47) |
| sAPPβ, pg/ml | 1.76 (2.23) | 1.94 (2.92) |
| BDNF, ng/ml | 28.1 (9) | 28.4 (7.5) |
| ApoA-1, mg/dl | 145.4 (44.5) | 151.7 (42.8) |
| TTR, mg/dl | 25.2 (5.2) | 26.1 (5.9) |
| C3(p), μg/ml | 10.8 (3.5) | 9.7 (4) |
| MDA-LDL, U/l | 106 (31) | 102 (31) |
| IL-6, pg/ml | 1.18 (1.91) | 1.17 (0.88) |
| IL-10, pg/ml | 2.08 (1.35) | 1.85 (0.87) |
| PSQI | 5.0 (3.0) | 4.0 (3.0) |
| MoCA-J | 26.0 (4.0) | 26.0 (3.0) |
| ADCS-MCI-ADL | 47.0 (5.5) | 47.0 (6.0) |
| MMSE-J | 28.0 (1.0) | 29.0 (1.0) |
| ADAS-Jcog | 3.7 (3.5) | 3.7 (2.3) |
| RBANS | 110.0 (17.0) | 110.0 (11.0) |
| CNS vital signs (Neurocognitive domain scores) | | |
| Social acuity | 5 (3) | 5 (3) |
| Reaction time | 868 (207) | 870 (239) |
| Complex attention | 12 (15) | 12 (15) |
| Cognitive flexibility | 24 (38) | 19 (29) |
| Executive function | 25 (37) | 21.5 (28) |
| Simple attention | 40 (1) | 40 (1) |

* mean ± SD

† median (IQR)

**Table 2. Brain atrophy and rCBF levels of participants in matcha and placebo groups at baseline.**

| Neuroimaging | Matcha group | Placebo group |
|---|---|---|
| | (n = 49) | (n = 50) |
| MRI | | |
| Severity of VOI atrophy | 0.59 (0.48)* | 0.79 (0.54) |
| Extent of VOI atrophy, % | 0.06 (2.14) | 0.74 (4.41) |
| Ratio of VOI/GM atrophy | 0.02 (1.25) | 0.25 (1.46) |
| SPECT | | |
| Severity | 0.93 (0.37) | 0.91 (0.62) |
| Extent, % | 3.53 (6.61) | 5.11 (10.39) |
| Ratio | 1.06 (1.75) | 1.37 (2.93) |

*median (IQR, Interquartile range)

**Table 3. Amyloid PET of participants in matcha and placebo groups at baseline.**

| Amyloid PET | Matcha group | Placebo group |
|---|---|---|
| | (n = 11) | (n = 13) |
| Baseline SUVR | 0.98 (0.06)* | 1.13 (0.24) |
| 12-month SUVR | 1.00 (0.10) | 1.06 (0.22) |

* median (IQR, Interquartile range)

months), group effect (matcha and placebo), and interaction between time and group (see **Methods** and **S1 File**). As shown in **Table 4**, sleep quality was measured using the PSQI, and cognitive function was assessed using the MMSE-J, MoCA-J, ADAS-Jcog, RBANS, and CNS vital signs.

The effect of match consumption on the outcomes was evaluated at 12 months. Social acuity in the neurocognitive domain scores of CNS vital signs showed significant improvement (difference; -1.39, 95%CI; -2.78, 0.002) ($P = 0.028$) and PSQI scores differed by 0.86 between the groups, indicating a trend of an improvement in sleep quality (95% CI; -0.002, 1.71) ($P = 0.088$) by 12-month matcha consumption (**Table 4**). Other cognitive functions including the primary outcomes (MoCA-J and ADCS-MCI-ADL) showed no significant changes in the matcha group compared with the placebo control group. We obtained results from both intent-to-treat and per-protocol analyses. The per-protocol analysis showed that differences in social acuity and PSQI scores between the groups were -1.44 (95% CI; -2.88, 0.001; $P = 0.077$) and 0.89 (95% CI; -0.001, 1.74; $P = 1.08$), respectively (**S3** and **S4** Tables).

Since the score for social acuity was assessed by the perception of emotion, we analyzed the outcomes of each cognitive test performed in CNS vital signs (**Fig 2**). Commission errors were

**Table 4. Effect of 12-month matcha intervention on sleep quality and cognitive functioning.**

| Outcome | Mixed effect model Baseline to 12 months | | | |
|---|---|---|---|---|
| | Estimate* | SE[†] | t-value | P-value |
| PSQI | 0.86 | 0.50 | 1.73 | 0.088[§] |
| ADCS-MCI-ADL | 0.05 | 0.82 | 0.06 | 0.949 |
| MoCA-J | 0.70 | 0.51 | 1.37 | 0.175 |
| MMSE-J | -0.38 | 0.33 | -1.18 | 0.242 |
| ADAS-Jcog | -0.33 | 0.44 | -0.75 | 0.458 |
| RBANS | -0.09 | 2.20 | -0.04 | 0.967 |
| CNS vital signs (Neurocognitive domain scores) | | | | |
| Social acuity | -1.39 | 0.62 | -2.23 | 0.028[‡] |
| Reaction time | 23.98 | 18.86 | 1.27 | 0.207 |
| Complex attention | 0.27 | 1.70 | 0.16 | 0.875 |
| Cognitive flexibility | 2.88 | 3.26 | 0.88 | 0.379 |
| Executive function | 2.49 | 3.20 | 0.78 | 0.438 |
| Simple attention | -1.17 | 0.87 | -1.35 | 0.181 |

* Estimate: The two-group difference in mean change from baseline to 12-momths calculated as described in the Methods. Positive value means higher value and negative value means lower value in matcha group comparing to placebo group, respectively.

[†] SE: Standard error

[‡] $P < 0.05$

[§] $P < 0.1$

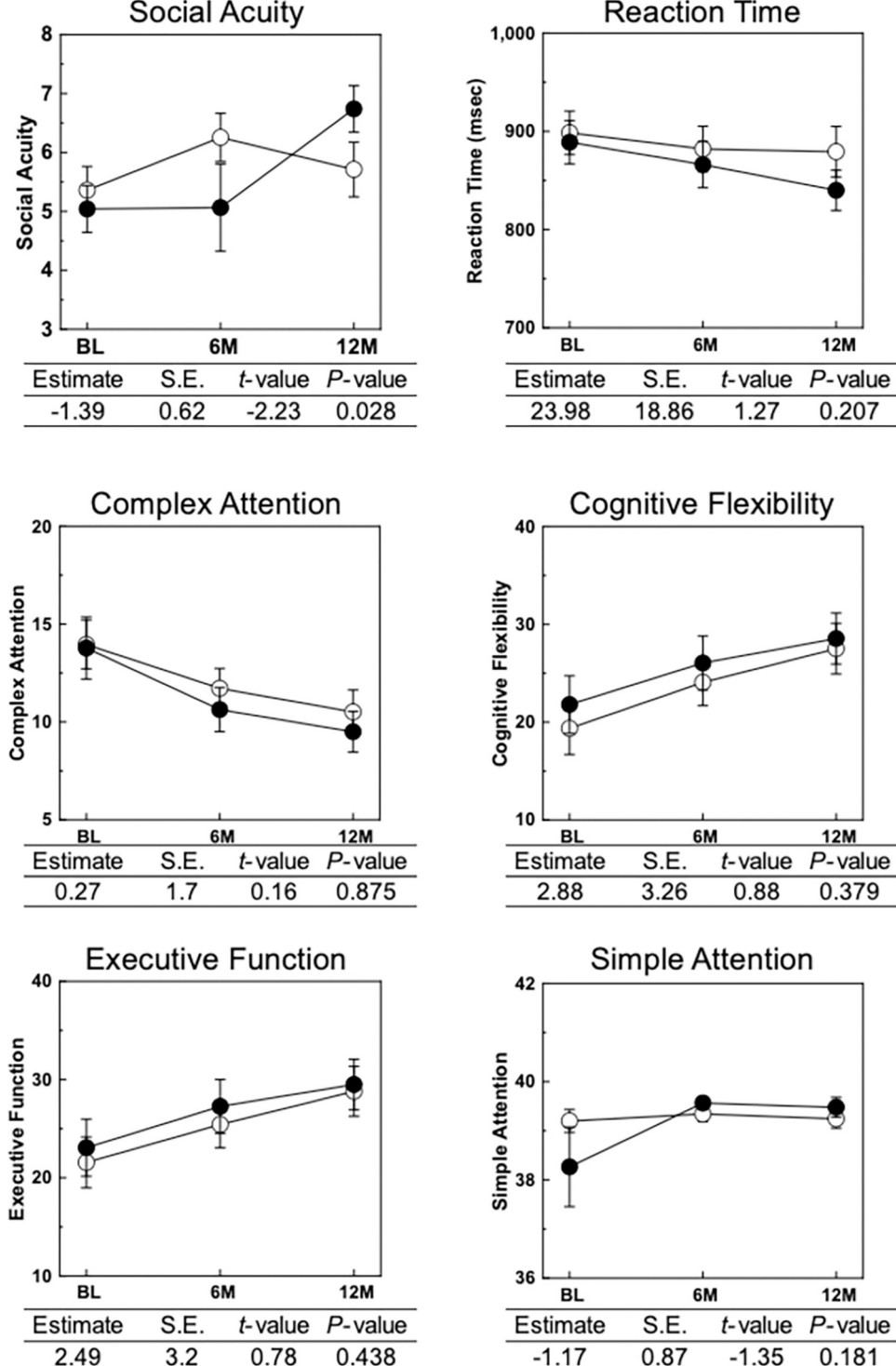

**Fig 2. Effect of 12-month matcha consumption on neurocognitive domain score in CNS vital signs.** The intervention effects on neurocognitive domain score in CNS vital signs are shown. Differences in the mean change from baseline to 12 months between the two groups were calculated using the mixed-effects model. Black circles represent the matcha group, and open circles represent the placebo group. The tables provided below represent the estimated differences between the groups; a positive value implies a higher value in the matcha group than in the placebo group, whereas a negative value suggests a lower value in the matcha group than in the placebo group.

reduced in the matcha group compared with the placebo group (difference; 1.16, 95% CI; 2.5E-03, 2.31) ($P$ = 0.035) in the perception of emotions. In the shifting attention test, the simple reaction time of the matcha group seemed to be shortened at 12 months compared with that at the baseline (difference; -48.93, 95% CI; -97.91, 0.06) ($P$ = 0.068) in the placebo group. In the Stroop test, the matcha group demonstrated a reduction in reaction time at 12 months compared with that in the baseline, whereas the placebo group exhibited virtually no change.

Although there was no significant improvement in cognitive function as evaluated by conventional cognitive functioning tests used in clinical diagnosis (Fig 3), 12-month matcha consumption improved social acuity, as assessed by the neurocognitive domain scores of CNS vital signs (Fig 2). Accordingly, we analyzed the association between social acuity and conventional cognitive functioning test scores in all participants and the stratified groups. At both baseline and 12 months, significant positive correlations were observed between the social acuity score and both MoCA-J and MMSE-J scores, whereas negative correlations were observed between the social acuity score and ADAS-Jcog score. No correlation was observed between social acuity and ADCS-MCI-ADL scores. Stronger correlations between the social acuity score and MoCA-J were observed in participants with SCD than in those with MCI and in the *APOE*4 noncarriers than in the *APOE*4 carriers. The social acuity scores of all participants were significantly correlated with the MoCA-J, MMSE-J, ADCS-MCI-ADL, ADAS-Jcog, and RBANS scores. However, in the MCI group, there was no significant correlation between social acuity and cognitive test scores. Conversely, in the SCD group, social acuity scores were significantly correlated with the MoCA-J, ADCS-MCI-ADL, and ADAS-Jcog scores.

### Neuroimaging biomarkers

The amyloid PET SUVR displayed no difference from the baseline and at 12 months in either group (Table 5). No significant changes were observed in hippocampal atrophy or rCBF levels after 12 months of matcha consumption.

### Discussion

In this study, unbiased randomization was conducted, and the trial was implemented as a triple-blind study with blinding for the intervention and placebo groups as well as the statistical analyst, thus enhancing the reliability of the study. In the current study, amyloid PET measurements using 18F-Florbetapir were performed in 24 (9 male and 15 female) participants who were *APOE*4 carriers. There were no biases in age, sex, *APOE* genotype, and amyloid burden between the intervention and non-intervention groups.

Present findings suggest that the consumption of matcha enhances certain cognitive functions, such as facial expression recognition and attention, and improves sleep quality, which are beneficial for maintaining cognitive function in older adult. Matcha consumption may be considered as a lifestyle improvement strategy for dementia prevention.

The PSQI total score indicated a trend toward improved sleep with matcha consumption. Despite the presence of caffeine, which disrupts sleep, matcha demonstrated a sleep-enhancing effect. This beneficial effect is attributed to theanine, a constituent of matcha. A crossover study by Hidese et al. on healthy participants with an average age of 48 years revealed that theanine intake (200 mg/day) for 4 weeks significantly reduced the total PSQI score [44]. As the change in the total score in the matcha group compared with that in the placebo group showed a decreasing trend ($P$ = 0.073), theanine was suggested to improve sleep quality. Furthermore, Ota et al. reported that in patients with schizophrenia with an average age of 42 years, theanine (250 mg/day) intake for 8 weeks alongside standard treatment led to a significant decrease in

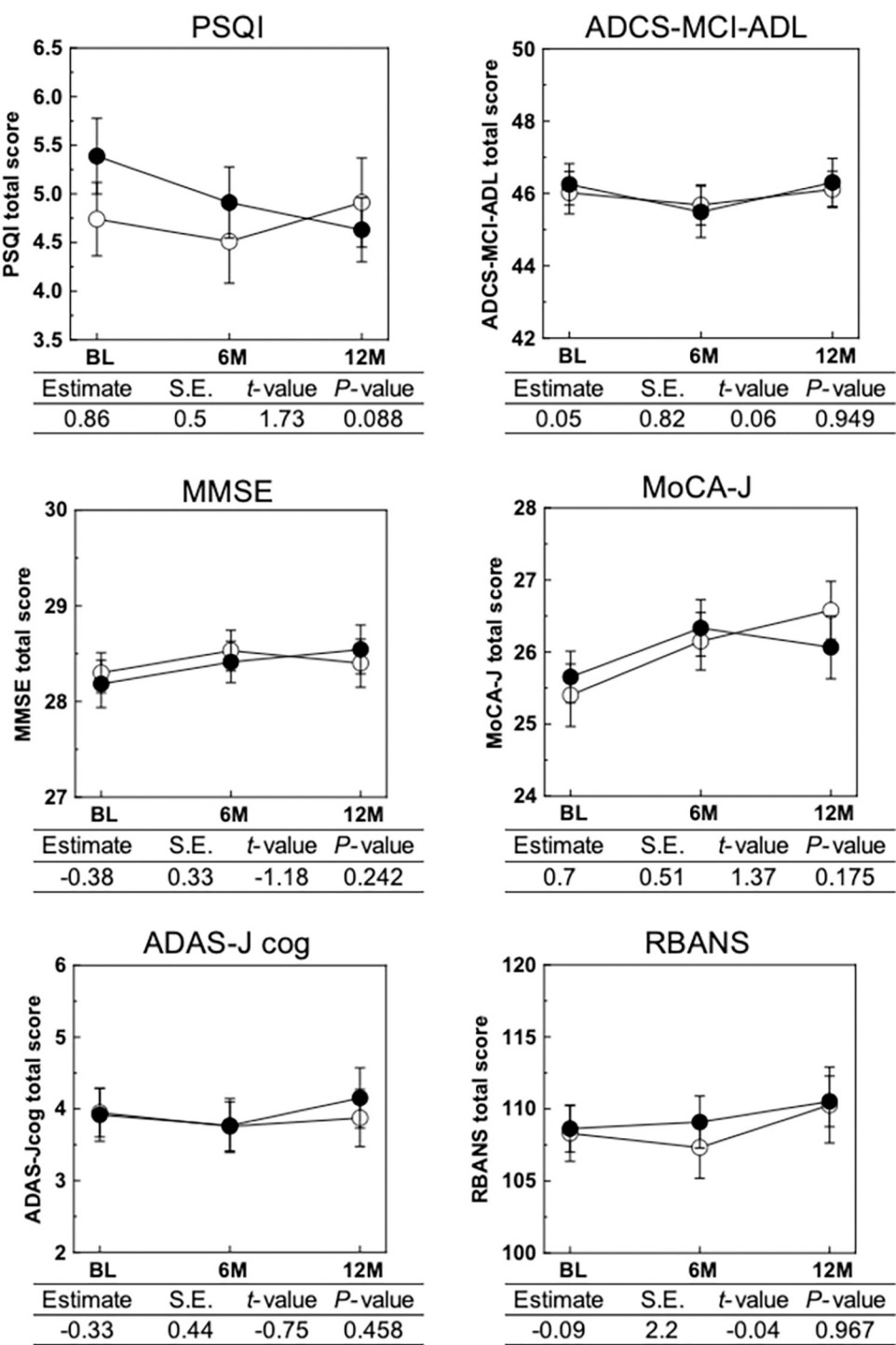

**Fig 3. Effect of 12-month matcha consumption on sleep quality and cognitive function.** The intervention effects on PSQI, ADCS-MCI-ADL, cognitive function assessed by MMSE-J, MoCA-J, ADAS-Jcog, and RBANS are shown. Differences in the mean change from baseline to 12 months between the two groups were calculated using the mixed-effects model. Black circles represent the matcha group, and open circles represent the placebo group. The tables provided below represent the estimated differences between the groups; a positive value implies a higher value in the matcha group than in the placebo group, whereas a negative value suggests a lower value in the matcha group than in the placebo group.

**Table 5. Effect of 12-month matcha intervention on neuroimaging.**

| Outcome | Mixed effect model Baseline to 12 months | | | |
|---|---|---|---|---|
| | Estimate* | SE† | t-value | P-value |
| Amyloid PET SUVR | 0.014 | 0.02 | 0.76 | 0.456 |
| MRI | | | | |
| Severity of VOI atrophy | 0.108 | 0.14 | 0.78 | 0.435 |
| Extent of VOI atrophy, % | -0.898 | 0.97 | -0.92 | 0.359 |
| Ratio of VOI/GM atrophy | 0.078 | 0.20 | 0.39 | 0.700 |
| SPECT | | | | |
| Severity | -0.001 | 0.04 | -0.02 | 0.982 |
| Extent, % | 0.177 | 0.97 | 0.18 | 0.855 |
| Ratio | -0.042 | 0.26 | -0.16 | 0.872 |

* Estimate: The two-group difference in mean change from baseline to 12-momths calculated as described in the Methods. Positive value means higher value and negative value means lower value in matcha group comparing to placebo group, respectively.

† SE: Standard error

total PSQI scores [45]. Thus, while theanine intake has been demonstrated to enhance sleep, this study provides the first report of the sleep-improving effects of matcha.

Substantial evidence suggests a close relationship between sleep and cognitive function [46–48]. Brachem et al. reported that in a longitudinal study of local residents aged 50–80 years, tracked for an average of 5.2 years, the risk of developing MCI was significantly increased (relative risk (RR) = 1.43, 95% CI; 1.12, 1.82) when the total PSQI score exceeded 5 points, as compared with those scoring 5 or less. Notably, in patients with MCI suffering from sleep disturbances (PSQI ≥ 5), significant improvement has been reported in the total PSQI and MoCA-J scores after 6 months of continuous positive airway pressure therapy in conjunction with donepezil administration [49]. These results suggest that enhancement of sleep quality could improve cognitive function.

In this study, a significant enhancement in social cognition was observed alongside an improvement in the total PSQI scores. It has been postulated that matcha may improve sleep quality and prevent or ameliorate cognitive decline. Thus, improvements in sleep quality associated with matcha intake may lead to cognitive function enhancement, even without the implementation of full-fledged sleep therapy. In the placebo group, a certain level of theanine was detected in the blood cells. This intervention study did not impose any restrictions on green tea intake before or during the study period. This may account for the presence of theanine, which may have been ingested during habitual green tea consumption.

Facial emotion recognition is impaired in patients with SCD and MCI [50, 51]. A cross-sectional analysis of 4,039 participants revealed that recognition of the emotion of facial expressions is affected at an early stage of cognitive impairment [52]. In the present study, a significant enhancement in social cognition was observed following matcha consumption. Social cognition is related to the perception of emotions, one of the tests of CNS vital signs, and the incidence of false responses was reduced by matcha consumption. False responses in the perception of emotions test pertained to inaccurate responses to facial expressions and descriptions (word meanings) administered during the test. A decrease in false response scores suggests that participants were more proficient in recognizing facial expressions and descriptions following 12-month matcha consumption. Social acuity, as evaluated by emotion recognition, was significantly associated with the MoCA-J score in patients with SCD and in *APOE*4

noncarriers, suggesting that this measure of social acuity may be a suitable test for the early stages of cognitive decline. However, the matcha intervention had no effect on the primary outcomes, MoCA-J and ADCS-MCI-ADL, possibly due to their sensitivity to slight declines in cognitive function. These tests are typically used during clinical diagnoses of cognitive impairment rather than for evaluating early-stage cognitive decline in the disease.

The matcha used in this study was powdered tea leaves. Its cultivation method differs from that of conventional green tea, as the tea leaves designated for matcha production are protected from sunlight prior to harvesting. This alternative cultivation approach increases the content of catechins and theanine, making it potentially superior to green tea in terms of the health benefits attributable to its ingredients. Participants drinking seven cups of matcha were excluded since it is equivalent to consuming 2 g of matcha contain 170 mg of catechins and 48 mg of theanine every day, which is the same amount of matcha as in this intervention study. In terms of catechin content, a single cup of matcha in a traditional tea ceremony, Tea Otemae, is equivalent to 250 mL (2–3 cups) of green tea; however, it should be noted that various types of green tea exist, and not all are directly comparable to matcha. Given the established health benefits of green tea and its constituents, we hypothesized that the increased catechin and theanine contents in matcha could help suppress cognitive decline and improve sleep quality.

The beneficial effects of matcha consumption revealed in this study may contribute to dementia prevention strategies, as these are relatively easy lifestyle changes to implement in daily life. Matcha consumption improved facial emotion recognition, as assessed by a computerized neurocognitive battery, since conventional psychological tests may not be suitable for evaluating cognitive decline in very mild cases. Thus, more sensitive neuropsychological testing, along with intervention strategies that are easy for elderly people to maintain, is necessary to assess the efficacy of these interventions in dementia prevention.

A limitation of the current study is the small number of participants. The observed PSQI score improvement within the intervention group suggests a potential influence of matcha consumption on sleep quality. However, future studies should employ more objective measures beyond the PSQI to comprehensively assess sleep architecture, including quantification of REM and non-REM sleep stages. A comprehensive clinical trial is required for a more robust evaluation of its effects on facial expression recognition. Furthermore, it is crucial to analyze the association between intervention outcomes and biomarkers to enhance our understanding of the mechanisms underlying the effects of matcha. Such investigations would make a significant contribution to the establishment of an effective ecosystem for dementia prevention.

## Conclusions

The findings of this long-term intervention study suggest that regular matcha consumption can enhance emotional perception and sleep quality in older adults experiencing cognitive decline.

## Supporting information

**S1 File. Supporting materials and methods.**
(PDF)

**S1 Table. Subsets of neurocognitive test corresponding to neurocognitive domain score used in the matcha intervention study.**
(PDF)

**S2 Table. Baseline clinical chemistry parameters of participants in the matcha and placebo groups.**
(PDF)

**S3 Table. Effect of 12-month matcha intervention on sleep quality and cognitive functioning in per-protocol analysis.**
(PDF)

**S4 Table. Effect of 12-month matcha intervention on neuroimaging in per-protocol analysis.**
(PDF)

# Acknowledgments

We thank all the participants in this study for their commitment and help in advancing this research. We thank Hitomi Ito and Makoto Inoue for helping to recruit the participants and Akiko Kaito for sample preparation. We also thank Tosie Shinagawa for critical reading of the manuscript. Registration number of present clinical research are UMIN000035658 (UMIN-CTR). The full trial protocol can be accessed in this clinical research registry. The data in this study can be accessed in Dryad Digital Repository (https://doi.org/10.5061/dryad.2280gb61r).

# Author Contributions

**Conceptualization:** Kazuhiko Uchida.

**Data curation:** Kohji Meno, Shan Liu, Hideaki Suzuki, Tatsuyuki Kakuma.

**Formal analysis:** Kohji Meno, Tatsumi Korenaga, Shan Liu, Hideaki Suzuki.

**Investigation:** Kohji Meno, Tatsumi Korenaga, Shan Liu, Yoshitake Baba, Chika Tagata, Yoshiharu Araki, Shuto Tsunemi, Kenta Aso, Shun Inagaki, Sae Nakagawa, Makoto Kobayashi, Miho Ota.

**Resources:** Tatsumi Korenaga, Hideaki Suzuki, Sae Nakagawa, Makoto Kobayashi, Takashi Asada, Miho Ota, Tetsuaki Arai.

**Supervision:** Tatsuyuki Kakuma, Takashi Asada, Takanobu Takihara, Tetsuaki Arai.

**Validation:** Tatsuyuki Kakuma, Takanobu Takihara, Tetsuaki Arai.

**Writing – original draft:** Kazuhiko Uchida, Kohji Meno.

**Writing – review & editing:** Kazuhiko Uchida, Takanobu Takihara.

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
