## [Decision Letter · Decision Letter 0]

10 Apr 2024

PONE-D-24-02571Effect of matcha green tea on cognitive functions and sleep quality in older adults with cognitive decline: A randomized controlled study over 12 monthsPLOS ONE

Dear Dr. Uchida,

Thank you for submitting your manuscript to PLOS ONE. After careful consideration, we feel that it has merit but does not fully meet PLOS ONE’s publication criteria as it currently stands. Therefore, we invite you to submit a revised version of the manuscript that addresses the points raised during the review process.

**The most pressing points to address are from Reviewer 2 who focused on the statistical aspects of the study and who raises several discrepancies from the CONSORT guidelines. Please ensure that you familiarise yourself with these criteria and address Reviewer 2's points in turn. Reviewer 1 and 3 raise some pertinent issues relating to definitions, language-use and areas that require expansion in the Introduction and Discussion sections. ** Please submit your revised manuscript by May 25 2024 11:59PM. If you will need more time than this to complete your revisions, please reply to this message or contact the journal office at plosone@plos.org. Please include the following items when submitting your revised manuscript:A rebuttal letter that responds to each point raised by the academic editor and reviewer(s). You should upload this letter as a separate file labeled 'Response to Reviewers'.A marked-up copy of your manuscript that highlights changes made to the original version. You should upload this as a separate file labeled 'Revised Manuscript with Track Changes'.An unmarked version of your revised paper without tracked changes. You should upload this as a separate file labeled 'Manuscript'.

We look forward to receiving your revised manuscript.

Kind regards,

Tamlyn Julie Watermeyer

Academic Editor

PLOS ONE

Journal Requirements:

Authors with competing interests

Enter competing interest details beginning with this statement:

I have read the journal's policy and the authors of this manuscript have the following competing interests: [Kazuhiko Uchida serves as a board member of MCBI Inc. Kohji Meno, Tatsumi Korenaga, Liu Shan, and Hideaki Suzuki are employees of MCBI Inc. Yoshitake Baba, Chika Tagata, Yoshiharu Araki, Shuto Tsunemi, Kenta Aso, Shun Inagaki, Sae Nakagawa, Makoto Kobayashi, and Takanobu Takihara are employees of ITO EN , LTD. This research received no external funding. ]

We note that one or more of the authors are employed by a commercial company. 

“The funder provided support in the form of salaries for authors, but did not have any additional role in the study design, data collection and analysis, decision to publish, or preparation of the manuscript. The specific roles of these authors are articulated in the ‘author contributions’ section.”

3. In the online submission form, you indicated that The data cannot be shared publicly without permission from all authors. Access to the data is available through the Institutional Data Access/Ethics Committee (contactable via the corresponding author) for researchers who meet the criteria for accessing confidential data.

Reviewers' comments:

Reviewer's Responses to Questions

**Comments to the Author**

1. Is the manuscript technically sound, and do the data support the conclusions?

Reviewer #1: Yes

Reviewer #2: Partly

Reviewer #3: Yes

2. Has the statistical analysis been performed appropriately and rigorously? 

Reviewer #1: Yes

Reviewer #2: I Don't Know

Reviewer #3: Yes

3. Have the authors made all data underlying the findings in their manuscript fully available?

Reviewer #1: No

Reviewer #2: Yes

Reviewer #3: Yes

4. Is the manuscript presented in an intelligible fashion and written in standard English?

Reviewer #1: Yes

Reviewer #2: Yes

Reviewer #3: Yes

5. Review Comments to the Author

**Reviewer #1: **This was a very interesting article. I have included some minor amendment below:

-Data statement – no explanation as to why data can’t be shared. States without permission from all authors but no explanation as to why all authors wouldn’t give permission. Is data owned by a third party?

- Typo on page 6, line 3, double comma

- Line 8 – could some context be given as to what is being classified as ‘middle age’ here.

- Within intervention section it would be useful to provide some justification as to why this dosage was decided upon.

- For statistical analysis is there justification as to why per protocol participants were analysed rather than Intent To Treat. CONSORT recommends reporting both in parallel, so some acknowledgement of ITT would be beneficial here, were the results the same?

- The results are interesting and placed well within previous literature. However there were no effects on primary outcomes and this should be acknowledged somewhere in the discussion.

**Reviewer #2:** Important note: This review pertains only to ‘statistical aspects’ of the study and so ‘clinical aspects’ [like medical importance, relevance of the study, ‘clinical significance and implication(s)’ of the whole study, etc.] are to be evaluated [should be assessed] separately/independently. Further please note that any ‘statistical review’ is generally done under the assumption that (such) study specific methodological [as well as execution] issues are perfectly taken care of by the investigator(s). This review is not an exception to that and so does not cover clinical aspects {however, seldom comments are made only if those issues are intimately / scientifically related & intermingle with ‘statistical aspects’ of the study}. Agreed that ‘statistical methods’ are used as just tools here, however, they are vital part of methodology [and so should be given due importance]. I look at the manuscript in/with statistical view point, other reviewer(s) look(s) at it with different angle so that in totality the review is very comprehensive. However, there should be efforts from authors side to improve (may be by taking clues from reviewer’s comments). Therefore, please do not limit the revision only (with respect) to comments made here.

COMMENTS: Kindly note that your ABSTRACT is well drafted (in my opinion), but is ‘assay type’. It is preferable [refer to item 1b of CONSORT checklist 2010: Structured summary of trial design, methods, results, and conclusions] to divide the ABSTRACT with small sections like ‘Objective(s)’, ‘Methods’, ‘Results’, ‘Conclusions’, etc. which is an accepted practice of most of the good/standard journals [including this one, though ‘The PLoS One Guidelines to Authors’ did not specify an Abstract format, it is desirable]. It will definitely be more informative then, I guess, whatever the article type may be. Since your ‘Article Type: Clinical Trial’, it very much desirable.

In addition, there are few issues/observations about which I have different opinion. Such observations/concerns are given below [if acceptable/agreed, try to incorporate them (consider positively wherever possible)]:

First is regarding choice of ‘Sample size’. One of the very important items in CONSORT guidelines is ’How sample size was determined (Item 7a)’ and expected/desired to described in ample details including all assumptions and software used. In fact, it is very surprising to note the ‘absence’ of the important term “CONSORT’’ despite that the manuscript is on ‘A randomized controlled clinical study/trial’. It is well-known that while reporting [findings from and even planning that is in ‘protocol’ of] ‘Clinical Trial’ one should follow/cover CONSORT guidelines.

In ‘Discussion’ it is just said/mentioned that “In this study, unbiased randomization was conducted, and the trial was implemented as a triple-blind study”. Regarding these important issues more discussion/detailed account is expected. Few important items {and all ‘Clinical Trials’ must follow} in CONSORT guidelines are ‘How sample size was determined (Item 7a)’,’ Random Sequence generation (Item 8a)’, ‘Allocation concealment (Item 9)’, Blinding (Item 11a). Since your article type is ‘Clinical Trial’, you are supposed to cover these items in the report or even in ‘Protocol’.

You may know that ‘Permuted Block Randomization’ ensures same group sizes (not simple randomization) and that randomization is a process [not only sequence generation] which includes ‘Allocation Concealment’}. Which ‘Randomization’ procedure is used in this study is not specified but almost equal group sizes are obtained. It must be ‘Permuted Block Randomization’ but such clear mention is desired.

In ‘Results-Participants’ section (as well as in Figure-1), you may use the word ‘screened’ in place of ‘recruited’ [As shown in Fig 1, we recruited 939 community-dwelling older adults aged 60–85 years and 124 participants were enrolled], in my opinion. May please confirm as word ‘recruited’ implies that 939 participated in trial. It may be known to you that when random allocation/assignment is used/done, any statistical comparison of baseline characteristics is not required. In this context [refer to Table-1], I request authors to read following note pasted from one famous standard textbook on ‘Medical Research Methodology’:

To provide a description of baseline characteristics is entirely reasonable (since it is clearly important in assessing to whom the results of the trial can be applied), however, statistical comparison of baseline characteristics when random allocation/assignment is used/done [often for good/standard/leading journals these days] is not required, because even if P-value(s) turn(s) out to be significant (while comparing baseline characteristics despite random allocation), it is, by definition, a false positive as you then are supposed to be testing ‘randomization’ then, which in any single trial may not balance all baseline characteristics (particularly when sample sizes are small). Remember that ‘randomization’ is a sort of ‘insurance’ and not a guarantee scheme. Authors may please refer to following articles:

References:

1. Stuart J. Pocock, et al., ‘Subgroup analysis, covariate adjustment and baseline comparisons in clinical trial reporting: current practice and problems’, Statistics in medicine, 2002; 21:2917–2930 [Particularly page 2927]

2. Harrington D, et al., ‘New guidelines for statistical reporting in the journal’, N Engl J Med 2019;381:285-6

[Important message (indirectly/ultimately indicated) from these articles: Never do any comparison with respect to ‘baseline’ characteristics {by applying statistical significance test(s)}, when allocation is done randomly].

However, Statistical comparison [only with respect to important/indicated variables] of baseline characteristics may be performed, to find out if analysis adjustment (say stratified analyses or else) is required with respect to these variables.

What is the purpose of Table-3? Why Between group comparison is performed separately at ‘Baseline’ and ’12-months’? What is/are interpretation of insignificant (both) P-vale(s)? Why not the comparison using ‘change scores’? In all the tables, test name is indicated only for few ‘variables/parameters’ {by giving symbol as superscript on P-value (last column)}. Is not it required to indicate test name in case of (i.e. with respect to) other/remaining variables/parameters? Or is it left for readers to guess? Remember/mind you that this is a scientific/academic document and so all details should be clearly/correctly communicated (do not take readers’ for granted). Most of the findings are reported without properly interpreting them or given relevance to study (example: page 26).

Only limitation of the study [A limitation of the current study is the small number of participants] mentioned on page 32 is agreed. However, does that mean {according to authors} there are none others? As pointed out in ‘important note’ above “This review pertains only to ‘statistical aspects’ of the study and so ‘clinical aspects’ should be assessed separately/independently. In my opinion, to make this article acceptable some amount of re-vision (re-drafting) may be needed. However, please do not limit the revision only (with respect) to comments made here. ‘Major revision’ is recommended.

**Reviewer #3**: The manuscript was well written and an interesting trial. I suggest some minor points to improve it:

- Introduction is well written with relevant links to supporting green tea literature. It would be useful to include rationale for the intervention timeframe (12 months). There does seem to be some literature on matcha tea interventions that haven't been included, which is more relevant than the included green tea literature - for example a paper by Sakurai (2020) used matcha for a 12 week intervention in elderly Japanese individuals.

- Page 7 line 5 - missing refernece for frequency of green tea consumption. It would also be useful here to have a brief explanation of this finding and the relation to this trial

- Page 7 line 8 - missing refernece for sleep disturbance/disorder stats, it would be more useful here to have the stats for sleep disorder prevalence in Japan rather than the US

- I am interested to know why age was stratified as >74 years old/ <74 years old, when the age range was 60-85 years. Is there a reason that 74 was chosen rather than 72? A brief comment at Page 9 line 2 would be useful

- Page 9 Line 16 - points 3 and 4 of the eligibility criteria are the same, please check

- Within eligibility criteria you mention that participants could not consume more that 7 cups of matcha per week, there is a comment within the discussion that the study did not impose restrictions on green tea intake during the trial, why was this decided? Did you collect information on habitual intake of green tea/matcha during the trial? If so, were there any differences in consumption between treatment groups? High consumption in the placebo group could explain lack of findings.

- Intervention information - I'd be interested to know what 2g of matcha equates to in comparison with consuming as a tea and/or how the catechin/theanine/caffeine content compares with green tea. I think this is particularly interesting given the citation of various green tea literature and the decision to exclude participants consuming 7+ cups matcha each week.

- Page 16 Line 2 - Please provide comment on if the repeated freezing/thawing of samples may have impacted the immunoassay results. Was this per protocol? It would also be useful here to include details of the blood sampling process. It would also be useful to include some information about the biomarkers assessed (e.g. assessing vitamin status)

- Page 23 Line 7 - compliance - earlier a pill count was mentioned, do you have the information on treatment compliance based on this? Was there a compliance cut off used for including participants in analysis?

- Within sample size calculation information I don't think it is clear why a medium-large effect size (cohen's d = 0.7) was used. Was this based off previous green tea literature? Please expand.

- Whilst this is recognised several times, consider highlighting each time that the PSQI finding is a trend only to be careful when over interpreting this finding.

I also found a few minor typos:

Page 6 Line 3 – extra comma

Page 20 line 4 – matcha misspelled

6. PLOS authors have the option to publish the peer review history of their article (what does this mean?). If published, this will include your full peer review and any attached files.

Reviewer #1: No

Reviewer #2: No

Reviewer #3: No

---

## [Author Response · Author response to Decision Letter 0]

26 May 2024

Reponse to the reviewers’ comments

Reviewer 1:

Thank you for your insightful suggestions, we have made several changes in the revised manuscript according to the Reviewer’s comments. These changes have enriched the context of our research, providing a clearer pathway for our readers to understand the implications of our findings.

1. Data statement – no explanation as to why data can’t be shared. States without permission from all authors but no explanation as to why all authors wouldn’t give permission. Is data owned by a third party?

Ans: After careful consideration and discussion among the authors, we decided to share the data in this study. The data has been uploaded to Dryad. 

https://doi.org/10.5061/dryad.2280gb61r

We have added the following sentence.

Page 36, lines 563-564

The data in this study can be accessed in Dryad Digital Repository (https://doi.org/10.5061/dryad.2280gb61r).

2. Typo on page 6, line 3, double comma

Page 5, lines 53

Ans: We have corrected it in the revised manuscript.

3. Line 8 – could some context be given as to what is being classified as ‘middle age’ here.

Ans: According to reference (5), people aged 45-54 years were classified as “middle age.” We changed the sentences as follows in the revised manuscript. 

Page 5, lines 58-60

Accumulating evidence suggests that lifestyle habits after middle age (approximately 45–54 years old) significantly impact the maintenance of cognitive function in older adults [3–5]

4. Within intervention section it would be useful to provide some justification as to why this dosage was decided upon.

Ans: As the amount of matcha in the traditional tea ceremony (Tea Otemae) is 2 g, a daily intake of matcha was set at 2 g.

We have added the following sentence.

Page 15, lines 233-235

As the amount of matcha in the traditional tea ceremony (Tea Otemae) is 2 g, the daily intake quantity of matcha was set at 2 g in this study.

5. For statistical analysis is there justification as to why per protocol participants were analysed rather than Intent To Treat (ITT). CONSORT recommends reporting both in parallel, so some acknowledgement of ITT would be beneficial here, were the results the same?

Ans: We have clarified the analytical methodologies concerning the intent-to-treat (ITT) and per-protocol (PP) in the revised manuscript. According to the ITT principle, we presented the data by ITT analysis in the original and revised manuscript. We analyzed changes in outcome variables from baseline to 12 months using a mixed-effects model using 46 participants in the matcha group and 45 in the placebo group. In addition to the ITT analysis, PP analysis was conducted, the results are shown in the S3 and S4 Tables of the revised manuscript, and corresponding modifications and textual additions have been made in the revised manuscript as follows:

Page 21, lines 350-352

A total of 99 participants were randomized and allocated to the matcha group (n = 49) and matched placebo group (n = 50).

Page 22, lines 366-373

The total number of completed questionnaires was 46 in the matcha group and 45 in the placebo group. During the intervention, each participant with MCI in the matcha and placebo groups was diagnosed with dementia at the 12-month evaluation. After excluding these two participants, the per-protocol analysis of the intervention effects included 45 and 44 participants in the matcha and placebo groups, respectively. According to the intention-to-treat principle, we analyzed the changes in outcome variables from baseline to 12 months were statistically analyzed using a mixed-effects model, with 49 and 50 participants in the matcha and placebo groups, respectively. 

Page 28, lines 422-425

We obtained results from both intent-to-treat and per-protocol analyses. The per-protocol analysis showed that differences in social acuity and PSQI scores between the groups were -1.44 (95% CI: -2.88, 0.001; P = 0.077) and 0.89 (95% CI: -0.001, 1.74; P = 1.08), respectively (S3 and S4 Tables).

6. The results are interesting and placed well within previous literature. However, there were no effects on primary outcome, and this should be acknowledged somewhere in the discussion.

Ans: We agree that the results of the primary outcome should be acknowledged. A corresponding statement has been added to the Discussion section. In the revised manuscript, we added the following sentences.

Page 28, lines 419-422

Other cognitive functions, including the primary outcomes (MoCA-J and ADCS-MCI-ADL), showed no significant changes in the matcha group compared with the placebo control group.

Page 34, lines 524-527

However, the matcha intervention had no effect on the primary outcomes (MoCA-J and ADCS-MCI-ADL) possibly due to their sensitivity to only slight declines in cognitive function. These tests are typically used during clinical diagnoses rather than for evaluating early-stage cognitive decline in the disease.

 

Reviewer 2:

Thank you for your crucial insights regarding the statistical elements of our study and the noted deviations from the CONSORT guidelines. These observations led us to meticulously revise the manuscript in accordance with the CONSORT standards. We believe that these adjustments have fortified the methods and results, and sharpened the conclusions, thereby enhancing the overall integrity of the manuscript.

1. Kindly note that your ABSTRACT is well drafted (in my opinion), but is ‘assay type’. It is preferable [refer to item 1b of CONSORT checklist 2010: Structured summary of trial design, methods, results, and conclusions] to divide the ABSTRACT with small sections like ‘Objective(s)’, ‘Methods’, ‘Results’, ‘Conclusions’, etc. which is an accepted practice of most of the good/standard journals [including this one, though ‘The PLoS One Guidelines to Authors’ did not specify an Abstract format, it is desirable]. It will definitely be more informative then, I guess, whatever the article type may be. Since your ‘Article Type: Clinical Trial’, it very much desirable.

Ans: Following the reviewer's comment, we have revised the abstract in accordance with the CONSORT guidelines

Pages 3-4, Abstract

Objective: Lifestyle habits after middle age significantly impact the maintenance of cognitive function in older adults. Nutritional intake is closely related to lifestyle habits; therefore, nutrition is a pivotal factor in the prevention of dementia in the preclinical stages. Matcha green tea powder (matcha), which contains epigallocatechin gallate, theanine, and caffeine, has beneficial effects on cognitive function and mood. We conducted a randomized, double-blind, placebo-controlled clinical study over 12 months to examine the effect of matcha on cognitive function and sleep quality. 

Methods: Ninety-nine participants, including 64 with subjective cognitive decline and 35 with mild cognitive impairment, were randomized, with 49 receiving 2 g of matcha and 50 receiving a placebo daily. Participants were stratified based on two factors: age at baseline and APOE genotype. Changes in cognitive function and sleep quality were analyzed using a mixed-effects model.

Results: Matcha consumption led to significant improvements in social acuity score (difference: -1.39, 95% confidence interval; -2.78, 0.002) (P = 0.028) as evaluated by the perception of facial emotions in cognitive function. The primary outcomes, that is, Montreal Cognitive Assessment and Alzheimer’s Disease Cooperative Study Activity of Daily Living scores, showed no significant changes with matcha intervention. Meanwhile, Pittsburgh Sleep Quality Index scores indicated a trend toward improvement with a difference of 0.86 (95% confidence interval; -0.002, 1.71) (P = 0.088) between the groups in changes from baseline to 12 months.

Conclusions: The present study suggests regular consumption of matcha could improve emotional perception and sleep quality in older adults with mild cognitive decline. Given the widespread availability and cultural acceptance of matcha green tea, incorporating it into the daily routine may offer a simple yet effective strategy for cognitive enhancement and dementia prevention.

2. In addition, there are few issues/observations about which I have different opinion. Such observations/concerns are given below [if acceptable/agreed, try to incorporate them (consider positively wherever possible)]:

First is regarding choice of ‘Sample size’. One of the very important items in CONSORT guidelines is ’How sample size was determined (Item 7a)’ and expected/desired to described in ample details including all assumptions and software used. 

In fact, it is very surprising to note the ‘absence’ of the important term “CONSORT’’ despite that the manuscript is on ‘A randomized controlled clinical study/trial’. It is well-known that while reporting [findings from and even planning that is in ‘protocol’ of] ‘Clinical Trial’ one should follow/cover CONSORT guidelines.

In ‘Discussion’ it is just said/mentioned that “In this study, unbiased randomization was conducted, and the trial was implemented as a triple-blind study”. Regarding these important issues more discussion/detailed account is expected. Few important items {and all ‘Clinical Trials’ must follow} in CONSORT guidelines are ‘How sample size was determined (Item 7a)’,’ Random Sequence generation (Item 8a)’, ‘Allocation concealment (Item 9)’, Blinding (Item 11a). Since your article type is ‘Clinical Trial’, you are supposed to cover these items in the report or even in ‘Protocol’.

You may know that ‘Permuted Block Randomization’ ensures same group sizes (not simple randomization) and that randomization is a process [not only sequence generation] which includes ‘Allocation Concealment’}. Which ‘Randomization’ procedure is used in this study is not specified but almost equal group sizes are obtained. It must be ‘Permuted Block Randomization’ but such clear mention is desired.

Ans: According to the reviewer’s comment, we added the following sentences and references in the Materials and Methods: “Study design,” “Randomization,” and “Participants” of the revised manuscript.

Page 8, line 110－page 9, line 123

Study design

A 12-month intervention was executed employing a randomized, double-blind, placebo-controlled, parallel-group design. Before the initiation of the study, the clinical trial was registered with the UMIN Clinical Trials Registry (UMIN-CTR), which is recognized by the International Committee of Medical Journal Editors (ICMJE) as an approved registry. This study was conducted as a randomized controlled trial, adhering strictly to the CONSORT guidelines. The required sample size was calculated using G*Power 3 software (Heinrich Heine University, Düsseldorf, Germany). We chose to assess the difference between the medians of two independent groups using the Wilcoxon or Mann–Whitney test. With the parameters set as two-tailed, logistic distribution, an effect size of 0.7, a significance level of 0.05, and a power of 0.8, it was determined that each group required 31 cases. Anticipating a dropout rate of 7 of 40 cases per group, we established a target sample size of at least 33 cases per group to validate the robustness of our findings (details are described in S1 File).

Page 9, lines 124-133

Randomization

Sequence generation

Eligible participants with SCD or MCI were randomized using a computer-generated sequence. Participants were stratified based on two stratification factors, age at baseline (≥ 74 years / < 74 years) and APOE genotype (ε4 positive/negative). Participants were stratified based on age at 74 years because a 10-year AD risk is higher in people ≥ 74 years old than in those < 74 years old [30]. Allocation to either the control or intervention group was intended to be randomized in a 1:1 ratio using computerized minimization, which aims to minimize differences between groups with respect to the stratification factors [31]. 

Page 9, line 134-page 11, line 166 

Allocation concealment mechanism

Participants underwent randomization using a dynamic program implemented via a Microsoft Excel macro, employing simple randomization rather than permuted block randomization. Upon running the program, the participants were assigned to either Group A or Group B. However, the program did not reveal which was the intervention or control group. The randomization table created by the program was retained until all baseline stratification factors were fully aligned and participant inclusion was complete, ensuring concealment of the allocation.

All eligible participants who provided informed consent and met the predefined inclusion criteria underwent randomization, a process administered at the discretion of the study site personnel overseeing recruitment and medical interviews. Subsequently, the study site transmitted a stratification factor response form to an independent allocation manager who was responsible for the allocation process, ensuring separation from the outcome evaluation analysis. The allocation manager utilized a dynamic randomization program incorporating provided stratification factors to assign participants to their respective groups based on the program's output. Physicians and nurses responsible for participant inclusion and symptom assessment remained blinded to group assignments.

The allocation sequence utilized a stratified approach, considering study site and medication compliance, employing a dynamic allocation method with the minimization technique. Concealment of randomization details from data management personnel and statistical analysts was upheld throughout the study while ensuring the database remained publicly accessible. The randomization list was securely maintained by designated allocation personnel throughout the study, mitigating any potential influence from principal investigators involved in the research.

Blinding

Throughout the study duration, participants were unaware of whether the capsules they consumed contained matcha or placebo until the study's conclusion was obtained. Physicians, nurses, and principal investigators at the study site were informed that the participants had been assigned to provisional groups (Groups A and B), yet remained uninformed about the specific allocation to matcha or placebo groups until the randomization list was released after the study period. No member of the research team knew the allocated sequence until the end of the statistical analysis after the database was locked.

Page 11, lines 167-170

Participants

Community-dwelling older adults aged 60–85 years were recruited and assessment for eligibility was performed at the University of Tsukuba Hospital (Tsukuba, Ibaraki, Japan) and Memory Clinic (Toride, Ibaraki, Japan) (Fig 1).

3. In ‘Results-Participants’ section (as well as in Figure-1), you may use the word ‘screened’ in place of ‘recruited’ [As shown in Fig 1, we recruited 939 community-dwelling older adults aged 60–85 years and 124 participants were enrolled], in my opinion. May please confirm as word ‘recruited’ implies that 939 participated in trial. It may be known to you that when random allocation/assignment is used/done, any statistical comparison of baseline characteristics is not required. In this context [refer to Table-1], I request authors to read following note pasted from one famous standard textbook on ‘Medical Research Methodology’.

Ans: I would like to thank you very much for your valuable comments and recommendation of the textbook. I think that this is the book by Joseph Abramson. 

I am sorry for the misuse of the term. We have corrected the word “screened” to “recruited.” We have also revised Fig 1 and Table 1.

4. To provide a description of baseline characteristics is entirely reasonable (since it is clearly important in assessing to whom the results of the trial can be applied), however, statistical comparison of baseline characteristics when random allocation/assignment is used/done [often for good/standard/leading journals these days] is not required, because even if P-value(s) turn(s) out to be significant (while comparing baseline characteristics despite random allocation), it is, by definition, a false positive as you then are supposed to be testing ‘randomization’ then, which in any single tri

---

## [Decision Letter · Decision Letter 1]

11 Jun 2024

PONE-D-24-02571R1Effect of matcha green tea on cognitive functions and sleep quality in older adults with cognitive decline: A randomized controlled study over 12 monthsPLOS ONE

Dear Dr. Uchida,

Thank you for submitting your manuscript to PLOS ONE. After careful consideration, we feel that it has merit but does not fully meet PLOS ONE’s publication criteria as it currently stands. Therefore, we invite you to submit a revised version of the manuscript that addresses the points raised during the review process.

We look forward to receiving your revised manuscript.

Kind regards,

Valerio Manippa

Academic Editor

PLOS ONE

Journal Requirements:

Reviewers' comments:

Reviewer's Responses to Questions

**Comments to the Author**

1. If the authors have adequately addressed your comments raised in a previous round of review and you feel that this manuscript is now acceptable for publication, you may indicate that here to bypass the “Comments to the Author” section, enter your conflict of interest statement in the “Confidential to Editor” section, and submit your "Accept" recommendation.

Reviewer #2: (No Response)

Reviewer #3: All comments have been addressed

2. Is the manuscript technically sound, and do the data support the conclusions?

Reviewer #2: (No Response)

Reviewer #3: Yes

3. Has the statistical analysis been performed appropriately and rigorously? 

Reviewer #2: (No Response)

Reviewer #3: Yes

4. Have the authors made all data underlying the findings in their manuscript fully available?

Reviewer #2: (No Response)

Reviewer #3: Yes

5. Is the manuscript presented in an intelligible fashion and written in standard English?

Reviewer #2: (No Response)

Reviewer #3: Yes

6. Review Comments to the Author

Reviewer #2: COMMENTS: Since all of the comments made on earlier draft are considered positively & mostly attended satisfactorily, I recommend the acceptance after minor revision. Although many modifications could be seen clearly, few details/modifications given while addressing comments were/are not very convincing [example: account regarding choice of ‘Sample size’]. ‘Minor revision’ is recommended so that authors can re-visit earlier comments and further modify by taking clues from earlier comments. However, please do not limit the revision only (with respect) to comments made there. More improvement is expected {note that ‘clinical implications’ are valuable}.

Reviewer #3: (No Response)

7. PLOS authors have the option to publish the peer review history of their article (what does this mean?). If published, this will include your full peer review and any attached files.

Reviewer #2: No

Reviewer #3: No

---

## [Author Response · Author response to Decision Letter 1]

20 Jun 2024

Response to the reviewer’s comments

Reviewer 2:

COMMENTS: 

Since all of the comments made on earlier draft are considered positively & mostly attended satisfactorily, I recommend the acceptance after minor revision. Although many modifications could be seen clearly, few details/modifications given while addressing comments were/are not very convincing [example: account regarding choice of ‘Sample size’]. 

‘Minor revision’ is recommended so that authors can re-visit earlier comments and further modify by taking clues from earlier comments. However, please do not limit the revision only (with respect) to comments made there. More improvement is expected {note that ‘clinical implications’ are valuable}.

ANSWER:

We are sincerely grateful for the opportunity to further improve our manuscript. In response to your valuable feedback, we have deepened the discussion regarding the clinical implications and limitations of our findings. We have incorporated a paragraph that explicitly outlines how our results concerning the effects of matcha consumption may contribute to dementia prevention strategies (page 35, line 554 – page 36, line 561). This paragraph also discusses the implications of our findings for the field of preventive medicine, thereby strengthening the connection between our research and its practical applications.

Regarding the chosen sample size, we acknowledge your feedback indicating that our rationale was not sufficiently articulated. To address this, we have anchored our determination of sample size in a robust statistical rationale, incorporating considerations such as power analysis and effect size, consistent with the guidelines recommended for studies of this nature. We have elaborated on this discussion in the revised manuscript, providing a more detailed justification and citing studies [30–32] employing analogous methodologies (page 8, line 120 – page 9, line 126).

We have diligently reviewed the manuscript to ensure consistency of content. Additionally, we have verified the references to ensure they are complete and correct, deleted duplicated references, and added three new references.

We added the following sentences and references in the Materials and Methods: 

Page 8, line 120 – Page 9, line 126

A meta-analysis was conducted to ascertain whether the effect size and statistical power were commensurate with the determined sample size, referencing three intervention studies that employed the Montreal Cognitive Assessment (MoCA) as the evaluative criterion [29–31]. Although the efficacy indices in this study vary slightly from those in previous research, we concluded that observing the anticipated effects outlined in the meta-analysis would render the data from all 66 patients a significant evidence base for preparing the subsequent validation study.

References

30. Tsolaki M, Kounti F, Agogiatou C, Poptsi E, Bakoglidou E, Zafeiropoulou M, et al. Effectiveness of nonpharmacological approaches in patients with mild cognitive impairment. Neurodegener Dis. 2011;8: 138–145.

31. Marzolini S, Oh P, McIlroy W, Brooks D. The effects of an aerobic and resistance exercise training program on cognition following stroke. Neurorehabil Neural Repair. 2013;27: 392–402.

32. Sukontapol C, Kemsen S, Chansirikarn S, Nakawiro D, Kuha O, Taemeeyapradit U. The effectiveness of a cognitive training program in people with mild cognitive impairment: A study in urban community. Asian J Psychiatr. 2018;35: 18–23.

Page 9, lines 132 – 134 

Anticipating a dropout rate of 7 of 40 recruited cases per group, we established a target sample size of at least 33 cases per group to validate the robustness of our findings (details are described in S1 File).

We added the following sentences in Discussion.

Page 35, line 554 – Page 36, line 561

The beneficial effects of matcha consumption revealed in this study may contribute to dementia prevention strategies, as these are relatively easy lifestyle changes to implement in daily life. Matcha consumption improved facial emotion recognition, as assessed by a computerized neurocognitive battery, since conventional psychological tests may not be suitable for evaluating cognitive decline in very mild cases. Thus, more sensitive neuropsychological testing, along with intervention strategies that are easy for elderly people to maintain, is necessary to assess the efficacy of these interventions in dementia prevention.

---

## [Decision Letter · Decision Letter 2]

9 Aug 2024

Effect of matcha green tea on cognitive functions and sleep quality in older adults with cognitive decline: A randomized controlled study over 12 months

PONE-D-24-02571R2

Dear Dr. Uchida,

We’re pleased to inform you that your manuscript has been judged scientifically suitable for publication and will be formally accepted for publication once it meets all outstanding technical requirements.

Kind regards,

Madepalli K. Lakshmana, Ph.D

Academic Editor

PLOS ONE

Additional Editor Comments (optional):

Reviewers' comments:

Reviewer's Responses to Questions

**Comments to the Author**

1. If the authors have adequately addressed your comments raised in a previous round of review and you feel that this manuscript is now acceptable for publication, you may indicate that here to bypass the “Comments to the Author” section, enter your conflict of interest statement in the “Confidential to Editor” section, and submit your "Accept" recommendation.

Reviewer #2: All comments have been addressed

Reviewer #3: All comments have been addressed

2. Is the manuscript technically sound, and do the data support the conclusions?

Reviewer #2: (No Response)

Reviewer #3: Yes

3. Has the statistical analysis been performed appropriately and rigorously? 

Reviewer #2: (No Response)

Reviewer #3: Yes

4. Have the authors made all data underlying the findings in their manuscript fully available?

Reviewer #2: (No Response)

Reviewer #3: Yes

5. Is the manuscript presented in an intelligible fashion and written in standard English?

Reviewer #2: (No Response)

Reviewer #3: Yes

6. Review Comments to the Author

Reviewer #2: COMMENTS: Further to what I stated earlier that all of the comments made on original draft were attended satisfactorily, I add that comment made on revised draft also is attended and I recommend the acceptance without any hesitation now.

Reviewer #3: (No Response)

7. PLOS authors have the option to publish the peer review history of their article (what does this mean?). If published, this will include your full peer review and any attached files.

Reviewer #2: No

Reviewer #3: No
